# HSP110 Regulates the Assembly of the SWI/SNF Complex

**DOI:** 10.3390/cells14110849

**Published:** 2025-06-05

**Authors:** Océane Pointeau, Manon Paccagnini, Natalia Borges-Bonan, Léo Biziorek, Sébastien Causse, Carmen Garrido, Laurence Dubrez

**Affiliations:** 1Institut National de la Santé et de la Recherche Médicale (Inserm), Inserm CTM UMR1231, Université Bourgogne-Europe, 21000 Dijon, France; oceane.pointeau@u-bourgogne.fr (O.P.); manon.paccagnini@u-bourgogne.fr (M.P.); natybonan@globo.com (N.B.-B.); leo.biziorek@ube.fr (L.B.); cgarrido@ube.fr (C.G.); 2Anticancer Center Georges François Leclerc-Unicancer, 21079 Dijon, France

**Keywords:** HSP110, chromatin remodeling complex, SWI/SNF

## Abstract

HSP110 is a ubiquitous chaperone contributing to proteostasis. It has a disaggregation activity and can refold denatured proteins. It can regulate fundamental signaling pathways involved in oncogenesis, such as Wnt/β-catenin, NF-κB and STAT3 signaling pathways. In gastric and colorectal cancer, HSP110 has been detected in the nucleus, and nuclear expression has been associated with the resistance of cells to 5-FU chemotherapy. Nuclear translocation of HSP110 is promoted by the exposure of cells to DNA-damaging agents. In a previous work, we demonstrated that nuclear HSP110 participates in the NHEJ DNA repair pathway by facilitating the recruitment of DNA-PKcs to Ku70/80 heterodimers at the site of DNA double-strand breaks. In the present work, analysis of HSP110s’ nuclear interactome revealed an enrichment of components from SWI/SNF chromatin remodeling complexes. We demonstrate that HSP110 is strongly associated with chromatin in temozolomide- and oxaliplatin-treated cells and directly interacts with the core subunit SMARCC2, thereby facilitating the assembly of SWI/SNF complexes. This work expands upon the role of HSP110, which regulates not only proteostasis but also the assembly of critical nuclear macromolecular complexes involved in the adaptive stress response.

## 1. Introduction

The chromatin organization in nucleosomes, where genomic DNA is tightly wrapped around histone octamers, ensures genome integrity by protecting DNA from DNA-damaging agents. Gene accessibility is controlled by chromatin-remodeling complexes. The chromatin remodeler switch/sucrose non-fermentable (SWI/SNF) complexes, also called BRG1 (Brahma homolog 1)/BRM (Brahma)—associated factor (BAF) complexes, are critical in the regulation of DNA accessibility for replication, transcription and DNA repair machineries. SWI/SNF complexes are evolutionarily conserved, megadalton multi-subunit protein complexes. In mammals, three main complexes have been described based on their subunit composition: canonical BAF (cBAF), polybromo-associated BAF (PBAF) and non-canonical BAF (ncBAF). They share the presence of one catalytic subunit, either SMARCA4 or SMARCA2 (also known as BRG1 or BRM, respectively), and the core subunits SMARCC2 and SMARCD1/2/3. Their specificity is related to the presence of specific accessory subunits. The different subunits constituting the complexes contain histone- or DNA-binding domains, generating different binding patterns and contact surfaces with chromatin [1,2]. Thanks to the ATPase activity of the catalytic subunit, SWI/SNF complexes can slide the nucleosome core to make DNA accessible to trans-acting factors. They are involved in the stress response since they can control the accessibility of DNA to DNA-repair machinery and they regulate the activity of the main heat shock transcription factor (HSF1) [3,4]. HSF is responsible for the heat shock response, i.e., the rapid and robust induction of heat shock proteins (HSPs) under exogenous or endogenous stress.

HSPs constitute a superfamily of molecular chaperones that control the conformation and stability of intracellular proteins. They assist in the folding of newly synthesized proteins, prevent the aggregation of denatured proteins and, for some of them, refold and disaggregate denatured proteins. Among the different HSPs, HSP110 is expressed in both the cytoplasm and the nucleus and undergoes nuclear translocation under hypoxia [5] or genotoxic stress conditions [6,7]. In the cytoplasm, HSP110 regulates fundamental signaling pathways, such as canonical apoptosis, Wnt/β-catenin, BCR/MyD88 and STAT3 signaling pathways [8,9,10,11]. Little is known about the nuclear functions of HSP110. In a previous work, we demonstrated that nuclear HSP110 participates in the non-homologous end joining (NHEJ) DNA repair pathway [6]. In the present work, we demonstrate that nuclear HSP110 can regulate the assembly of SWI/SNF complexes. HSP110 can directly interact with the core subunit SMARCC2 and facilitate the recruitment of the SMARCA4 catalytic subunit.

## 2. Materials and Methods

### 2.1. Cell Culture and Treatment

SW480, a human adenocarcinoma cell line, and HEK 293T, human embryonic kidney cells, were cultured in DMEM culture medium (Dulbecco’s Modified Eagle Medium #L0104-500; Dominique Dutscher, Bernolsheim, France) supplemented with 10% *v*/*v* fetal calf serum (Dominique Dutscher #500105C1C) and a mixture of penicillin (100 U/mL), streptomycin (0.1 mg/mL) and amphotericin B (0.25 µg/mL) (PAN Biotech GmBH, Aidenbach, Germany; #P06-07300). SW480 cells were treated with 200 µM Temozolomide (Sigma-Aldrich, Saint Louis, MO, USA; dissolved in DMSO #PHR1437) or 50 µM Oxaliplatin (Accord healthcare France, Lille, France). The ATM/ATR inhibitor ETP 46464 (Clinisciences, Nanterre, France #HY-15521, stock concentration: 5 mM in DMSO) and SMARCA4 inhibitor (Clinisciences #A19059-1, stock concentration: 5 mM in DMSO) were used at 5 µM and 500 nM, respectively.

### 2.2. Plasmid and siRNA Transfection

pCMV5-SMARCC2 (BAF170)-Flag and pCMV5-SMARCA4 (BRGI)-Flag were a gift from Joan Massague (Addgene, Cambridge, MA, USA; plasmid # 19142 and # 19143, respectively), and pcDNA-HSP110-Flag was described in [6]. The siRNAs used were HSP110-targeting siRNA (Thermo Fisher, Waltham, MA, USA; #S21242), SMACC2-targeting siRNA (Thermo Fisher, #S13150) and SMACA4-targeting siRNA (Thermo Fisher, #S13130). SW480 cells were transfected with Lipofectamine RNAiMAX^®^ (Thermo Fisher, #13778075) in 25 cm^2^ culture flasks for siRNAs and with JetPeI (Ozyme, Saint-Cyr l’Ecole, France; #POL101000053) for plasmid DNA.

### 2.3. Total Cell Extraction

SW480 cells were lysed in RIPA buffer (150 mM NaCl, 1% Triton, 0.5% sodium deoxycholate, 1% SDS, 50 mM Tris-HCl (pH 8) supplemented with complete protease inhibitor cocktail (Merck, Rahway, NJ, USA; #11697498001) and phosphatase inhibitors (Cocktail 2, Sigma-Aldrich, Saint Louis, MO, USA, #P5726). Proteins were quantified and denatured prior to Western blot analysis.

### 2.4. Cytoplasm–Nucleus Fractionation

SW480 cells were lysed on ice using the NE-PER™ Nuclear and Cytoplasmic Extraction Kit (Thermo Fisher, #78833), according to the manufacturer’s instructions. All extraction buffers were supplemented with complete protease inhibitor cocktail (Merck #11697498001). Proteins were then quantified and subjected to Western blot analysis.

### 2.5. Salt Gradient Chromatin Fractionation

Cells were lysed in CEBN buffer (2 mM HEPES (pH 7.8), 2 mM KCl, 0.3 mM MgCl_2_, 68 mM sucrose, 10% glycerol, 0.2% NP-40) supplemented with complete protease inhibitor cocktail (Merck, Rahway, NJ, USA; #11697498001) for 10 min on ice and then subjected to centrifugation at 16,000× *g* for 5 min at 4 °C. The supernatant containing the cytoplasmic fraction was collected. The pellet was lysed in nuclear buffer soluble (0.3 mM EDTA and 0.02 mM EGTA) supplemented with complete protease inhibitor cocktail (Merck #11697498001) for 30 min on ice and then subjected to centrifugation at 16,000× *g* for 5 min at 4 °C. The supernatant corresponding to the nucleoplasm fraction was collected. The remaining pellet containing chromatin-bound proteins was incubated with low salt extraction buffer (5 mM Tris (pH 8), 0.005% NP-40, 0.1 M NaCl) supplemented with complete protease inhibitor cocktail (Merck #11697498001), and the samples were centrifuged at 16,000× *g* for 5 min. The resulting supernatant was collected as the chromatin-bound protein fraction corresponding to that specific salt concentration. The pellet was then resuspended in the next buffer with a higher salt concentration (0.3 M to 0.7 M NaCl), and the process was repeated. Proteins were quantified and denatured prior to Western blot analysis.

### 2.6. Immunoprecipitation and Pull-Down Assay

FLAG-tagged SMARCA4, SMARCC2 and HSP110 proteins were transiently expressed in HEK 293 T or SW480 cells by plasmid transfection. Forty-eight hours post-transfection, the cells were lysed in lysis buffer (lysis buffer A) (Cell Signaling Technology, Danvers, MA, USA; #9803) for 30 min. Immunoprecipitation was performed using magnetic agarose beads conjugated with anti-FLAG antibody (Anti-DYKDDDDK, Thermo Fisher #A36797). For each condition, 50 µL of anti-FLAG beads was incubated with 200 µg of total protein in a final volume of 500 µL of lysis buffer. The samples were incubated at 4 °C for 2 h under gentle rotation. Following incubation, the beads were washed twice with 500 µL of lysis buffer to remove non-specifically bound proteins. The bound proteins were eluted directly by adding 40 µL of Laemmli buffer (63 mM Tris HCl (pH 6.8); 20% Glycerol, 2% SDS, 0.01% bromophenol blue and 10 mM Dithiothreitol (DTT)) to the beads. The eluted samples were heated twice at 95 °C, with an intermediate cooling step on ice, and then analyzed by Western blot.

For the pull-down experiments, SMARCC2-Flag or SMARCA4-Flag was transiently expressed in HEK 293 T cells for 24 h. The cells were lysed in lysis buffer (lysis buffer B) (50 mM Tris HCl (pH 7.4), 200 mM NaCl, 5 mM EDTA, 5% Glycerol, 1% NP-40, supplemented with complete protease inhibitor cocktail (Merck #11697498001)), and 50 µg of total protein was incubated with 50 µL of anti-FLAG magnetic beads for 1 h at 4 °C under gentle rotation. The beads were washed with buffer A and used immediately for the interaction assay. Immobilized SMARCC2-Flag or SMARCA4-Flag proteins were incubated with recombinant Human Hsp105/HSP110 protein (Abcam, Cambridge, MA, USA; #Ab78790) or cell lysate from SW480 cells transfected with scramble, HSP110, SMARCC2 or SMARCA4 siRNAs. The mixture was incubated for 2 h at 4 °C with gentle rotation. The beads were then washed twice with lysis buffer A. The bound proteins were analyzed by Western blotting.

### 2.7. Western Blotting

Protein concentrations were determined using the DC Protein Assay kit (Bio-Rad, Hercules, CA, USA; #5000116). Equal amounts of protein (20–50 µg per lane) were separated by SDS-PAGE on an 8% polyacrylamide gel in migration buffer (25 mM Tris Base, 190 mM glycine, 0.1% SDS). Proteins were transferred onto polyvinylidene difluoride (PVDF) membranes (0.45 µm pore size) using Towbin transfer buffer containing 10% ethanol (28 mM Tris Base, 213 mM glycine) for 1 h at 100 V. Following transfer, the membranes were blocked for at least 1 h at room temperature in TBST buffer (20 mM Tris Base, 150 mM NaCl, 0.1% Tween-20) containing 5% bovine serum albumin (BSA). The membranes were then incubated with primary antibodies recognizing the proteins of interest. An appropriate secondary antibody conjugated to horseradish peroxidase was used for detection, except for the β-actin protein, which is already coupled to HRP. The list and conditions of use of the antibodies are shown in Appendix A. Protein bands were visualized using a chemiluminescence detection kit (Clarity™ Western ECL Substrat BioRad #170–5061 or SuperSignal™ West Femto Thermo Fisher #34095). All images were captured using BioRad Chemidoc and analyzed using Image Lab 6.0 software (Bio-Rad, Hercules, CA, USA). The expression levels of the proteins of interest were normalized to those of protein references, such as β-actin, Lamin B1, GAPDH and TBP.

### 2.8. Immunofluorescence and Proximity Ligation Assays

Cells grown on No. 1.5 coverslips were fixed using 4% PFA in phosphate-buffered saline (diluted from 16% formaldehyde solution (*w*/*v*) methanol Free Thermo Fisher #28908) for 10 min at room temperature. The cells were washed once in TBS, permeabilized for 10 min with 100% methanol at −20 °C and washed again in TBS, 0.1% *v*/*v* Tween-20. The samples were blocked using TBS, 0.1% *v*/*v* Tween-20, 3% *w*/*v* bovine serum albumin (BSA) for 1 h at room temperature and then incubated overnight with primary antibodies at 4 °C in a humidified environment (1:100 dilution). The samples were washed four times with TBS 0.1% *v*/*v* Tween-20, 15 min per wash, at room temperature. For immunofluorescence, the samples were incubated with the secondary antibody (Invitrogen, Waltham, MA, USA) anti-rabbit Alexa488 (#A11034) for 1 h at room temperature, washed four times in TBS 0.1% *v*/*v* Tween-20, rinsed in water and mounted on a microscopy slide using DAPI containing Prolong Gold antifade reagent (Life Technologies, Waltham, MA, USA).

Alternatively, for PLA, the procedure recommended by DuoLink was carried out using anti-rabbit plus (Sigma-Aldrich #DUO92002) and anti-mouse minus (Sigma-Aldrich #DUO92004) PLA probes and PLA detection kit orange (Sigma-Aldrich #DUO92007).

### 2.9. Microscopy and Image Analysis

Microscopy images were acquired using a Zeiss Axio Imager 2 (Zeiss GmBH, Oberkochen, Germany) equipped with ×20 0.46 NA, ×40 0.6 NA and ×63 1.45 NA Plan Apochromatic objectives and an mRM4 CCD camera. Illumination was provided by an HXP120 metal halide lamp (Zeiss GmBH). The filter sets used were 49 HE (DAPI #488049-0000) and 38 HE (Alexa488 #489038-0000). Image analysis was carried out using the ICY software (http://icy.bioimageanalysis.org (accessed on 24 April 2023)) on 10 randomly taken fields (based on DAPI). Nuclei were segmented using the Active Contour plugin on DAPI signal; PLA and HSP110 foci were detected using the Spot Detector plugin v1.9.2.0. Fluorescence signals were measured in maximal projections of unmodified images.

### 2.10. Mass Spectrometry Analysis

Cells were lysed on ice for 20 min in IP buffer (10 mM Tris (pH 8), 50 mM NaCl, 10 mM EDTA (pH 8), 0.05% Triton, 0.5% desoxycholate, 0.5% SDS, 1 mM NaVO4, 1 mM NaF) supplemented with complete protease inhibitor cocktail (Merck #11697498001). The lysates were centrifuged at 16,000× *g* for 10 min at 4 °C, and protein concentrations in the supernatants were measured using a modified Lowry assay (Bio-Rad). Proteins were incubated overnight at 4 °C with either anti-HSP110 antibody (Abcam #Ab109624) or control IgG (Cell Signaling Technology, Danvers, MA, USA; #2729). A 1:1 mix of protein A and protein G agarose beads (50 µL; Millipore, Rahway, NJ, USA) was used to pull down the immunocomplexes for 30 min at room temperature. The beads were then washed three times with IP buffer. The immunoprecipitated proteins were solubilized with 2× Laemmli buffer (50 mM Tris pH 6.8, 8% *v*/*v* glycerol, 1.6% *w*/*v* SDS, 0.8% *v*/*v* 2-mercaptoethanol, bromophenol blue), heated at 95 °C for 30 min and processed by SDS-PAGE (Biorad, 10% acrylamide, 50 µL). The quantity of loaded proteins was adjusted in order to obtain a comparable amount of immunoprecipitated HSP110 in the untreated and treated samples. The gel was colored with Coomassie blue, and the proteins were reduced with TECP solution, alkyled with IAA and digested in gel by trypsin at 37 °C overnight. Peptides were extracted with 60% acetonitrile (ACN), 0.1% formic acid (FA) and then with 100% ACN. Peptides were purified with C18 microspin columns (Havard Apparatus) and suspended in 30 µL 2% ACN, 0.1% FA.

ESI-TRAP mass spectrometry was performed by the CLIPP platform using nanoUPLC (nanoRSLC, Thermo Fisher) coupled to a mass spectrometer equipped with an Advion TriVersa Nanomate nanospray source and composed of an ion trap in tandem with an orbitrap (LTQ-Orbitrap Elite, Thermo Scientific). A total of 3 to 7 μL of sample was loaded onto an enrichment pre-column (Acclaim PepMap C18 75 μm × 20 mm; Thermo Fisher) with solvent A (2% acetonitrile, 0.1% formic acid in water) at a flow rate of 5 μL/min for 3 min. The peptides were then eluted by increasing the concentration of solvent B (80% acetonitrile, 0.1% formic acid) from 2% to 25% in 160 min on a separation column (AcclaimPepMap 75 μm × 250 mm, 2 μm; Thermo Fisher) maintained at 33 °C, at a flow rate of 300 nL/min. The Mascot search algorithm (v25.1) was used to identify proteins from MS and MS/MS data. The search was performed in the protein database restricted to the ‘Human’ taxonomy (Uniprot, 16 May 2019). The MS/MS data obtained with the Orbitrap were then validated with ProlineStudio software (v2.0.1). The results were validated with an FDR of 1% on MMPs and proteins, with a protein score greater than or equal to 25 and a peptide length of at least 6 amino acids.

### 2.11. Statistics

Statistical analysis was performed with GraphPad Prism 6 for Windows (San Diego, CA, USA) using the Mann–Whitney test for multiple comparisons. Statistical significance was set at *p* < 0.05. All summary results are presented as the mean ± SEM.

## 3. Results

### 3.1. Proteome Analysis of Nuclear HSP110

To extend our investigation of the nuclear functions of HSP110, we analyzed the HSP110–nuclear interactome by ESI-TRAP mass spectrometry. The nuclear translocation of HSP110 was induced by treating cells with the alkylating agent temozolomide at 200 µM (TZM), which induced the phosphorylation of ATR within 24 h of treatment (Figure 1a). We observed a rapid induction of HSP110 protein expression in both the cytoplasm and the nucleus. While the cytoplasmic expression of HSP110 decreased after 24 h, the nuclear expression remained high (Figure 1b,c). HSP110 was immunoprecipitated from the nuclear enriched fraction in the untreated and TMZ-treated cells (Figure 1d), and HSP110 partners were identified by mass spectrometric analysis. As expected, the number of hits was higher in the treated cell samples than in the untreated cells (Figure 1e, Table 1). We detected HSP70, in accord with the well-reported function of HSP110 as an HSP70 cofactor, and DNA-PKcs, in accord with the published role of HSP110 in the NHEJ DNA repair pathway [6]. Histones H2A, H4 and H3.1; several RNA-binding proteins; the transcription factors DDX9 and eEF-1α; and different GTPases were also detected. We found a strong enrichment of components of the chromatin remodeler SWI/SNF complexes. Five subunits of theses complexes were detected with a higher probability of interaction in the treated cells. They include the ATPase catalytic subunit SMARCA4, the core subunits SMARCC2 and SMARCB1 and the accessory proteins SMARCE1 and PBRM1 (Table 1).

### 3.2. HSP110 Is Recruited into the ATP-Dependent Chromatin-Remodeling SWI–SNF Complex

Salt gradient fractionation of chromatin demonstrated that TMZ (Figure 2a,b) and oxaliplatin (Figure 2c) treatment induced a strong enrichment of HSP110 in the chromatin-enriched fractions. As a control, SMARCC2, which is an important structural subunit of the SWI/SNF complex, and histone H3 were detected in the chromatin-enriched fraction in both control and DNA-damaged cells. Inhibiting ATR did not interfere with the association of SMARCC2 with chromatin but prevented the recruitment of HSP110 to chromatin (Figure 2d), suggesting that this event depends on DNA damage. On the other hand, inhibition of SMARCA4 did not affect the recruitment of HSP110 to the chromatin, demonstrating a SWI/SNF complex activity-independent process. Altogether, these results suggest that TMZ- or oxaliplatin-induced DNA damage promotes the association of HSP110 with chromatin.

### 3.3. HSP110 Can Directly Bind SMARCC2

We confirmed that SMARCC2 and SMARCA4 are partners of HSP110 in co-precipitation experiments. SMARCA4 and SMARCC2 can co-precipitate with HSP110 (Figure 3a), and inversely, HSP110 can co-precipitate with SMARCA4 and SMARCC2 (Figure 3b). However, while recombinant human HSP110 can directly bind SMARCC2, it cannot bind to SMARCA4 (Figure 3c). The PLA assay demonstrated a strong proximity of endogenous HSP110 with SMARCC2 in oxaliplatin-treated SW480 cells (Figure 3d). Silencing of SMARCA4 did not modify the quantity of HSP110 that binds to the chromatin, confirming the SMARCA4-independent mechanism observed in Figure 2d. However, silencing of SMARCC2 decreased the enrichment of HSP110 into chromatin-bound fractions (Figure 3e), suggesting that the recruitment of HSP110 into the chromatin at least partly depends on SMARCC2.

### 3.4. HSP110 Is Important for the Assembly of the SWI/SNF Complex

In the current model for the assembly of SWI/SNF complexes, the first step is a dimerization of SMARCC2, followed by the recruitment of core and accessory subunits and, finally, the ATPase SMARCA4 [12]. We investigated the ability of HSP110 to modulate the assembly of the complex by assessing the SMARCC2:SMARCA4 association. Flag-SMARCC2 was immobilized on Flag-beads and then incubated with cell lysates from cells in which HSP110 or SMARCA4 had been silenced. Figure 4a demonstrates that the downregulation of SMARCA4 did not affect the binding of HSP110 to SMARCC2. Inversely, silencing of HSP110 decreased the binding of SMARCA4 to SMARCC2 (Figure 4a). The PLA experiments confirmed the importance of HSP110 for the SWI/SNF complex assembly in DNA-damaged cells. Indeed, silencing of HSP110 blocked the DNA-damage-induced SMARCA4:SMARCC2 proximity (Figure 4b,c).

## 4. Discussion

Although many studies described the presence of HSP110 in the nucleus [5,6,7,13,14], its nuclear functions remain poorly documented. HSP110 has been detected in a large panel of gastric tumor samples and has been associated with a bad prognosis and high risk of recurrence after chemotherapy [14]. A deletion mutation in the HSP110-encoding gene (mutation HSP110T17) has been detected in more than 95% of colon cancer cells displaying a microsatellite instability (MSI) phenotype. This mutant encodes a truncated form of HSP110 (HSP110DE9) that acts as a dominant negative and completely inhibits HSP110 nuclear translocation and activity [15]. The tumoral expression of this mutant was correlated with a good response to 5-fluorouracil and was proposed as a biomarker of response to this chemotherapy [15,16]. Accordingly, depletion or HSP110DE9-mediated inactivation of HSP110 sensitizes cancer cells to oxaliplatin and 5-FU [14,15,17,18]. A nuclear accumulation of HSP110 was observed in response to cell exposure to DNA-damaging agents, such as adriamycin [7], oxaliplatin [6] or TMZ (present).

Our mass spectrometry analysis detected approximately 40 potential HSP110 nuclear partners. The presence of histone H3.1 and the transcription factor eEF-1α in the nuclear HSP110 interactome is consistent with Wheat et al.’s analysis of the HSP110 interaction network [19]. We detected HSP70, which is often found in complex with HSP110. The two proteins cooperate to refold, disaggregate and reactivate misfolded substrates [20]. Many of the hits identified were also found to be potential HSP70 partners [21]. These include nucleophosmin; chromobox family member 1 (CFA); polybromo 1 (PBRM1); Tubulin Folding Cofactor A (ACF/TBCA); histones H3.1 and H4; nucleolin; DNA-dependent protein kinase catalytic subunit (DNA-PKcs); Poly(A) binding protein cytoplasmic 1 (ABP1); heterogeneous nuclear ribonucleoprotein (hnRNP)-A1, -H, -M and -K; RNA-binding protein 14 (RNA-BP14); Rab-6A; Ran and KRas; and the SWI/SNF complex subunits SMARCA4, SMARCC2, SMARCB1 and SMARCE1.

We detected a strong enrichment of subunits from chromatin remodeling complexes and histones H2A, H4 and H3.1 in TMZ-treated cells. Accordingly, we demonstrate that HSP110 is strongly associated with chromatin in TMZ- and oxaliplatin-treated cells in an ATR/ATM- and SMARCC2-dependent manner.

SMARCC2 is a core subunit of SWI/SNF complexes, which are composed of up to 15 subunits. The modular organization and assembly of SWI/SNF complexes were deciphered using complementary biophysical, biochemical, molecular biology and genetic analyses [12]. A SMARCC2 dimer forms the basis of a scaffold, which subsequently incorporates other core subunits, including SMARCD1/2/3 and SMARCB1, followed by the accessory subunits. The catalytic subunit SMARCA4 or SMARCA2 is recruited in the final stage of assembly, completing the formation of the active SWI/SNF complexes [12]. Little is known about the mechanisms regulating the assembly of the SWN/SNF complex. We demonstrate that the chaperone HSP110 directly interacts with SMARCC2. HSP110 is critical in the assembly of the SWI/SNF complex, as evaluated by SMAARC2–SMARCA4 proximity in DNA-damaged cells.

The role of HSP proteins in the proper assembly and stability of protein complexes has been well demonstrated, particularly for HSP90. It interacts with the Particle for Arrangement of Quaternary structure (PAQosome), whose function is to stabilize the quaternary structure of macromolecular complexes [22]. HSP90 cooperates with the PAQosome in regulating the assembly, stability and function of snoRNP (small nucleaolar RNA protein complexes) involved in the maturation of ribosomal RNA, spliceosome, RNA-polymerase complexes, phosphatidylinositol 3-kinase-related kinases (PIKKs) and the MRN complex involved in HR (homologous recombination) DSB (double-strand breaks) DNA repair signaling pathway [22]. In the MRN complex, HSP90 is essential for the expression and stabilization of some important subunits, in the assembly of the complex and in its recruitment to DSBs [22,23].

HSP110 has also been associated with the assembly of protein complexes ensuring the repair of DNA DSBs, which are particularly deleterious to cells. The NHEJ DNA repair pathway involves the DNA double-strand breaks (DSBs) sensors ku70/80, the DNA-dependent protein kinase catalytic subunit (DNA-PKcs), ligases and cofactors assembled in a macromolecular complex. HSP110 can directly interact with the Ku70/80 heterodimer and facilitate the recruitment of DNA-PKcs [6]. Accordingly, our proteome analysis (Table 1) identified DNA-PKcs as an HSP110 partner. In colorectal cancer cells, nuclear HSP110 colocalizes with the DNA DSBs marker γH2AX [6]. SWI/SNF complexes have been implicated in DNA DSBs repair, including NHEJ, by relaxing the chromatin structure and allowing for exposure of damaged DNA fragments to repair machinery [24]. As with HSP110, mutation or deletion of SWI/SNF subunits leads to increased sensitivity to DNA-damaging agents [25,26]. The SWI/SNF complex is rapidly recruited to the site of DNA breaks, but the involved mechanisms are not clearly determined. Most studies focused on the recruitment of the catalytic subunit SMARCA4 to DSBs, but its activity and whether its recruitment was associated with the assembly of SWI/SNF complexes were not determined. SWI/SNF subunits may recognize histone acetylation marks [27,28]. In addition to phosphorylating H2AX at the DSB sites, ATM and ATR can also phosphorylate SMARCC2 and SMARCA4, which favors their interaction with DSB-associated proteins BRIT1/MCPH1 and Rb and the E2F1 and E4F1 transcription factors [29,30,31,32]. SMARCC2 is regulated by other post-translational modifications. The RIPK1 kinase can directly interact and phosphorylate SMARCC2 and enhance the nucleosome remodeling activity of SWI/SNF complexes in vitro [33]. The methylation of lysine residues of SMARCC1/2 by Lysine-specific histone demethylase 1A (LSD1) is critical to maintaining the integrity of SWI/SNF complexes, since demethylated SMARCC1 is rapidly targeted for proteasome-mediated degradation [34]. Abundant data in the literature demonstrate that HSPs can favor or interfere with certain protein modifications. For instance, HSP110 has been shown to favor the phosphorylation of STAT3 [9]. Our data here demonstrate that SWI/SNF assembly can be regulated by HSP110. Whether HSP110s’ role in the regulation of DNA repair pathways involves the SWI/SNF complex and the modulation of post-translational modifications remains an important question to address.

Altogether, HSP110 appears as a potent regulator of genomic stability in cells exposed to genotoxic stress. It accumulates into the nucleus, where it promotes chromatin relaxation by facilitating SWI/SNF assembly and stimulates DNA repair through the NHEJ pathway. A role for SWI/SNF complexes in heat shock response has been well established, especially in Yeast [3,35,36]. HSF1 is the master stress response transcription factor that binds to heat shock response elements found in the promoters of HSP family member-encoding genes, inducing their rapid transcription under stress. SMARCA4 can interact with HSF1 and facilitate its recruitment to heat shock response elements [4,37]. The recruitment of SWI/SNF complexes to the promoter region is critical for preparing chromatin and allowing for the binding of cofactors and transcription machinery required for HSP expression during heat shock [3,36]. The role of HSP110 in the assembly and recruitment of SWI/SNF complexes to HSP promoters remains to be determined. As demonstrated for HSP90 [22], HSP110 could also be a potent regulator of the dynamic association, stability and activity of macromolecular complexes. Thus, in addition to their function as molecular chaperones involved in regulating the conformation and stability of intracellular proteins, HSPs also appear to be guardians of the assembly and activity of macromolecular complexes. Both HSP90 and HSP110 cooperate with HSP70 in their activity. Further investigation is required into the role of HSP70, the mechanism of HSP110-mediated regulation of the assembly of the complexes and the dynamic and kinetic aspects of its intervention.

## Figures and Tables

**Figure 1 cells-14-00849-f001:**
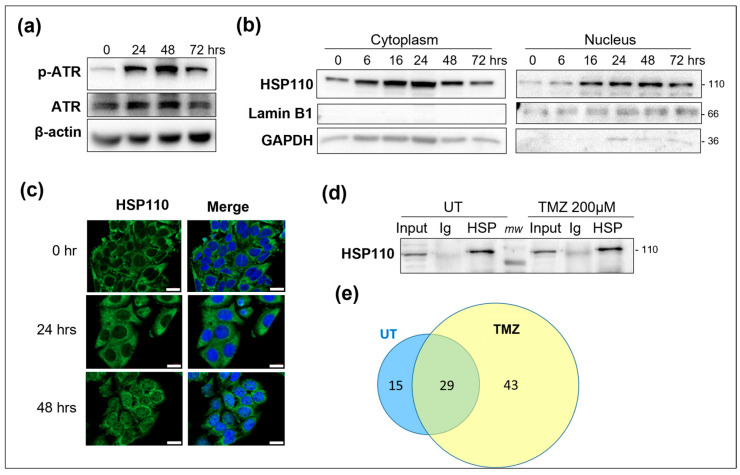
SW480 colon cancer cells were exposed to 200 µM TMZ for the indicated (**a**–**c**) hours or 48 h (**d**,**e**). (**a**) Western blot analysis of the phosphorylation of ATR. β-actin was used as a loading control. One representative experiment is shown, *n* = 3. (**b**) Modulation of HSP110 expression in nuclear and cytoplasmic-enriched fractions. Lamin B1 and GAPDH were used as nuclear and cytoplasmic markers, respectively. One representative experiment is shown, *n* = 2. (**c**) Immunofluorescence analysis of nuclear translocation of HSP110. Nuclei were stained by DAPI, Scale bar=10µM. *n* = 3. (**d**) Immunoprecipitation of HSP110 in nuclear-enriched fractions from control (UT) or TMZ-treated cells. The quantity of protein loaded was adjusted in order to obtain comparable HSP110 amounts in UT and treated cells. (**e**) Venn diagram of nuclear HSP110 protein partners identified by mass spectrometry.

**Figure 2 cells-14-00849-f002:**
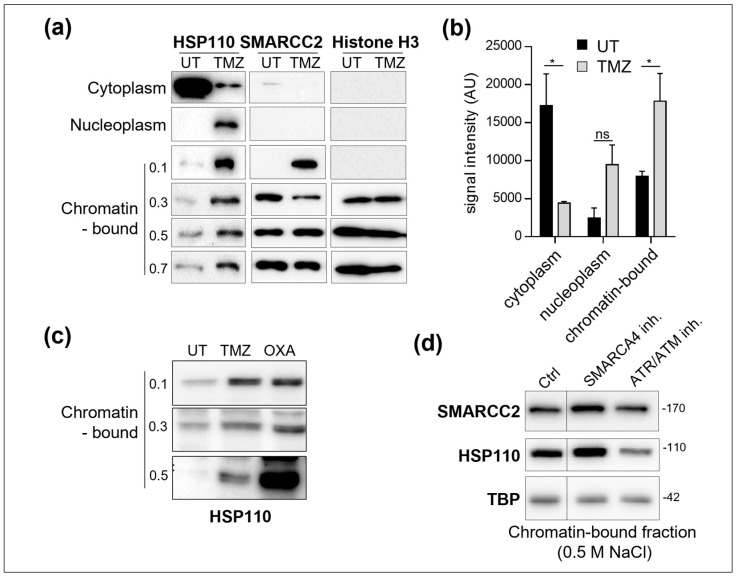
SW480 colon cancer cells were exposed to 200 µM temozolomide (TMZ) (**a**–**d**), 50 µM oxaliplatin (OXA) (**c**) or vehicle (UT) for 48 h. Western blot analysis of HSP110, SMARCC2 and Histone H3 in cytoplasm, nucleoplasm and chromatin-bound enriched fractions. Histone H3 is used as a loading control. Proteins were extracted from chromatin by using increasing concentrations of NaCl (0.1–0.7 M). (**b**) Quantification of the Western blot shown in (**a**). Mean ± SEM, *n* = 4. *: *p*-value < 0.05, Mann–Whitney analysis. (**d**) SW480 colon cancer cells were exposed to 50 µM oxaliplatin (OXA) in the presence of 500 nM SMARCC4 inhibitor-1 or 5 µM ETP46464 (ATR/ATM inh.). Western blot analysis of SMARCC2 and HSP110 in chromatin-bound fraction extracted with 0.5 M NaCl. TBP (TATA-binding protein) is used as a loading control.

**Figure 3 cells-14-00849-f003:**
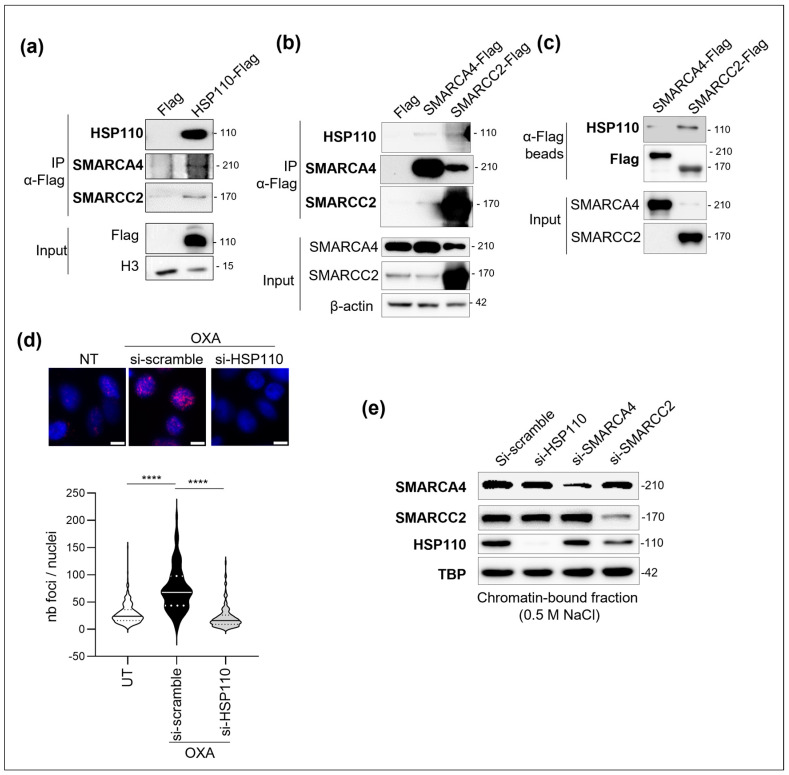
(**a**,**b**) Immunoprecipitation analysis of the interaction of HSP110 with SMARCC2 and SMARCA4. HSP110-Flag, SMARCA4-Flag or SMARCC2-Flag was expressed in HEK293 cells and immunoprecipitated. Western blot analysis of HSP110, SMARCA4 and SMARCC2. One representative experiment is shown. (**c**) SMARCA4-Flag or SMARCC2-Flag was expressed in HEK293 cells, immobilized on anti-flag beads and incubated with HSP110 recombinant protein. Western blot analysis of HSP110, Flag, SMARCA4 and SMARCC2. One representative experiment is shown. (**d**) Top panel: representative image of PLA between HSP110 and SMARCC2 in SW480 cells treated with 50 µM oxaliplatin (OXA) or vehicle (UT) for 48 h. PLA controls in HSP110 expression silenced cells is provided. Scale bar=10µM. Lower: quantification of PLA signal (at least 75 cells per condition). Mann–Whitney analysis, ****: *p* < 0.0001. (**e**) SW480 colon cancer cells transfected with scramble, HSP110, SMARCC2 or SMARCA4 siRNAs were exposed to 50 µM oxaliplatin for 48 h. Western blot analysis of SMARCC2 and HSP110 in chromatin-bound fraction extracted with 0.5 M NaCl. TBP (TATA-binding protein) is used as a loading control.

**Figure 4 cells-14-00849-f004:**
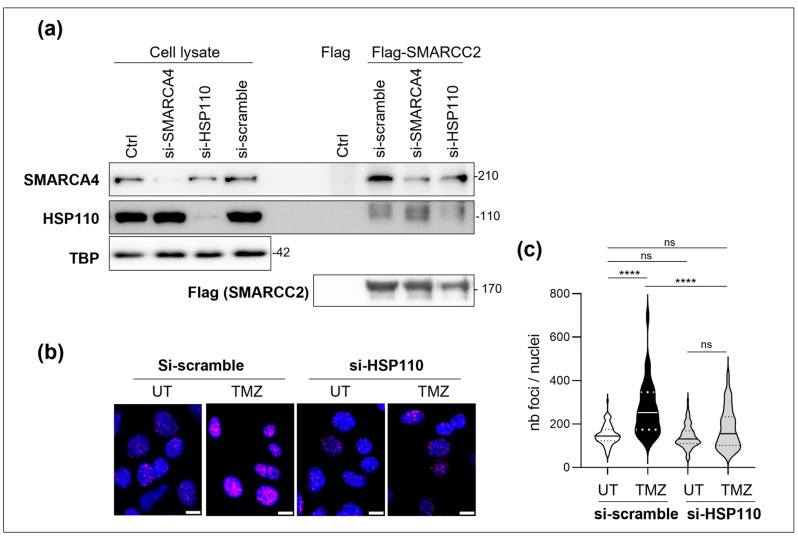
(**a**) Flag-SMARCC2 proteins were immobilized on anti-flag beads and incubated with cell extracts from SW480 cells transfected with scramble, HSP110 or SMARCA4 siRNAs. TBP (TATA-binding protein) is used as a loading control. (**b**) Representative image of PLA between SMARCC2 and SMARCA4 in SW480 cells transfected with scramble or HSP110 siRNA and treated or not (UT) with 200 µM temozolomide (TMZ) for 48 h. Scale bar = 10 µM. (**c**) quantification of PLA signal shown in panel (**b**) (at least 75 cells per condition). Mann–Whitney analysis, ****: *p* < 0.0001; ns: not significant.

**Table 1 cells-14-00849-t001:** Proteomic analysis of HSP110-interacting proteins in nuclear fraction from untreated (UT) or TMZ-treated SW480 cells.

Functional Sub-Category	Protein	UniProtKB	Protein Set Score		Function
			UT	TMZ	
Chaperones	Nucleophosmin	P06748	42.82	80.31	
	HSP70	P17066	-	**105.01**	
	CFA/TBCA	O75347	-	34.14	Tubulin-specific chaperone
Chromatin remodeling complexes:					
SWI/SNC complexes	SMARCA4/BRG1	P51532	35.37	**108.16**	**ATPase**
	SMARCC2/BAF170	Q8TAQ2	39.94	**211.08**	**Core subunit**
	SMARCB1/SNF5	Q12824	-	**34.6**	**Core subunit**
	SMARCE1/BAF57	Q969G3	39.41	**47.91**	Accessory proteins
	PBRM1/BAF180	Q86U86	129.83	**278.66**	Accessory proteins with
					bromo domain
ISWI complex	ACF1/BAZ1A	Q9NRL2	39.41	47.91	Accessory proteins with bromo domain
Histones	H2A	P04908	-	43.25	
	H4	P62805	-	141.71	
	H3.1	P68431	-	40.9	
DNA repair	DNA-PKcs/DNPK1	P78527	-	43.34	NHEJ pathway
RNA-binding proteins	FXR1		36.17	-	
	ABP1	P11940	34.46	42.6	
	Nucleolin	P19338	84.39	181.27	
	hnRNP A2	P09651	42.8	106.34	
	hnRNP H	P31943	46.31	88.47	
	hnRNP A0	Q13151	45.26	38.54	
	hnRNP M	P52272	-	84.4	
	hnRNP K	P61978	-	35.01	
	hnRNP U		-	241.27	
	hnRNP D0		-	35.39	
	αCP3		-	42.52	
	Sam68		-	45.49	
	H3.1		-	49.31	
	COAA or RNA-	Q96PK6	-	184.8	
	BP14				
Transcriptional regulators	DDX9		49.94	146.86	DNA-dependent RNA
					helicase, TF
	eEF-1α		-	54.38	
	Prohibitin-2		-	36.03	
GTPases and regulators	Rab-3D, 6A, 15 or 33B	O95716, P20340, P59191, Q9H082,	33.06	-	Ras-related protein Rab
	ASAP2/PAP	O43150	38,77	40.31	
	GNAT3 or GNA12	A8MTJ3 or Q03113	-	39.75	
	Rac2	P15153	-	51.97	
	Ran	P62826	-	56.9G	
	Gα12		-	39.75	
	KRas	A0A024RAV5	-	52.76	

Protein set score was obtained with the algorithm Mascot: The higher the score is, the higher is the probability of interaction. Not highlighted: protein score set <50; Light grey highlighted: 50 < protein score set <100; Dark grey highlighted: protein score set >100.

## Data Availability

The original contributions presented in this study are included in the article/Appendix A. Further inquiries can be directed to the corresponding author.

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
