# Peer review of "HSP110 Regulates the Assembly of the SWI/SNF Complex"

_cells, 2025, doi:10.3390/cells14110849_

Round 1
Reviewer 1 Report
Comments and Suggestions for Authors
This work continues the past work on that nuclear HSP110 participates in the NHEJ DNA repair pathway by facilitating the recruitment DNA-PKcs to Ku70/80 heterodimer at the site of DNA double strand breaks, to further clarify the more detail mechanism on HSP110 interacting with SMARCC2 and then facilitating the assembly of SWI/SNF complexes especially in te-mozolomide and oxaliplatin-treated cells. Although the proposed finding cannot fully answer the whole mechanism of the binding and assembly process for the SWI/SNF complexes, at least by using several indirect evidences this study has demonstrated HSP110 really has strong relationship to the assembly process. The molecular biological works are very solid and persuasive, and the finding is also very interesting and pave the way to unveil more detail mechanism and pathway for the assembly of SWI/SNF complexes. I feel the current manuscript can be accepted as its current form for publication.
Author Response
Comment: This work continues the past work on that nuclear HSP110 participates in the NHEJ DNA repair pathway by facilitating the recruitment DNA-PKcs to Ku70/80 heterodimer at the site of DNA double strand breaks, to further clarify the more detail mechanism on HSP110 interacting with SMARCC2 and then facilitating the assembly of SWI/SNF complexes especially in te-mozolomide and oxaliplatin-treated cells. Although the proposed finding cannot fully answer the whole mechanism of the binding and assembly process for the SWI/SNF complexes, at least by using several indirect evidences this study has demonstrated HSP110 really has strong relationship to the assembly process. The molecular biological works are very solid and persuasive, and the finding is also very interesting and pave the way to unveil more detail mechanism and pathway for the assembly of SWI/SNF complexes. I feel the current manuscript can be accepted as its current form for publication.
Reply: Thank you very much for your analysis and your positive evaluation of our work.
Reviewer 2 Report
Comments and Suggestions for Authors
In this manuscript, O. Pointeau and colleagues investigate the nuclear functions of HSP110, a molecular chaperone involved in maintaining the folding and structural stability of proteins. While the cytoplasmic functions of Hsp110 are relatively well characterized, little is known about its modus operandi in the nucleus. The study proposes proteomics, biochemical and cellular analyses suggesting that when translocated in the nucleus, HSP110 interacts with chromatin, and help the assembly of SMARCC2 with SMARCA4, two proteins of the SWI/SNF complex responsible for chromatin remodeling upon DNA damages. On the whole, the proposed experiments appear to be well designed and the conclusions drawn are in good agreement with their results. One can only regret that the study does not go further in proposing precise molecular functions for HSP110 – but investigating the latter represents a substantial amount of work, that might be the scope of another manuscript.
Major comments
The abstract presents the results obtained, but could be improved by explicitly proposing their significance and underline the impact of this study.
The actual number of replicates performed is missing for many of the experiments presented here. This should be stated in order to convince the reader that the results obtained can be trusted. For instance, the legend of Figure 1 ends with the sentence: “one representative experiment is shown”: over how many repeats? Do the authors refer to all of the panels and experiments, or a single one? Please clarify.
Figure 1: in panel 1b, the authors show an increase of nuclear Hsp110 upon TMZ treatment. Yet in panel 1d, while the total amount of Hsp110 remains constant between TMZ treated and untreated cells, the amount of immunopurified nuclear Hsp110 appears very similar between both conditions. If this discrepancy is due to the quantities loaded on the gel for WB experiments in Panel 1d, the authors should state so in their manuscript. Otherwise, how do the authors explain this observation?
Proteomics analysis: please indicate the number of replicates. If the experiment was only performed once, it can only be considered as preliminary. Or at least, some of the proteomics results presented in Table 1 (differential enrichment of various proteins in TMZ treated/untreated IPs) should be validated by western blot analyses. The current good practice is to perform triplicates/quadriplicates of IPs (biological replicates, not technical ones), and interpret them in volcano plots showing fold change of enrichment vs. p-value. In addition, the proteomics data must be deposited to a recognised database such as the ProteomeXchange Consortium / PRIDE. In Table 1, what do the different grey levels correspond to? Please add the color-coding meaning to the legend for more clarity.
Figure 2 a and c: Hsp110 WB lines corresponding to UT and TMZ treatments (chromatin-bound fractions) display significantly different profiles between panel in figure 2a and 2c. For instance, in panel 2a, the Hsp110 signals in UT condition increase between fraction 0.1 and 0.5, while it decreases in panel 2c. Could the authors propose an explanation about this observation? Is it because of a significant variability between replicates?
Minor comments:
- Material and methods section, Mass spectrometry analyses paragraph l. 172, please correct the first sentence: “HSP110 from nuclear-enriched fractions of control and temozolomide-treated cells, cells were lysed on ice for 20 minutes in IP buffer (1X) “
- Strange presentation of Table 1 (words are randomly split into 2 consecutive lines. For example, the last n of nucleophosmin is on the next line
Discussion:
- With this study, Hsp110 appears to play a role of quaternary chaperone allowing the assembly of the SWI/SNF complex. In various cell types and compartments, this quaternary chaperone function is also performed by other molecular machines such as the PAQosome (RuvBL1/2 associated with Hsp70/90 and other helper proteins) – perhaps the authors should draw a parallel between Hsp110 and the latter?
- 356 “(…)but its’ activity, and whether its’ recruitment(…)”: both “its” do not use an apostrophe
Author Response
In this manuscript, O. Pointeau and colleagues investigate the nuclear functions of HSP110, a molecular chaperone involved in maintaining the folding and structural stability of proteins. While the cytoplasmic functions of Hsp110 are relatively well characterized, little is known about its modus operandi in the nucleus. The study proposes proteomics, biochemical and cellular analyses suggesting that when translocated in the nucleus, HSP110 interacts with chromatin, and help the assembly of SMARCC2 with SMARCA4, two proteins of the SWI/SNF complex responsible for chromatin remodeling upon DNA damages. On the whole, the proposed experiments appear to be well designed and the conclusions drawn are in good agreement with their results. One can only regret that the study does not go further in proposing precise molecular functions for HSP110 – but investigating the latter represents a substantial amount of work, that might be the scope of another manuscript.
Major comments
Comment 1: The abstract presents the results obtained, but could be improved by explicitly proposing their significance and underline the impact of this study.
Reponse 1: We completed the abstract.
Comment 2: The actual number of replicates performed is missing for many of the experiments presented here. This should be stated in order to convince the reader that the results obtained can be trusted. For instance, the legend of Figure 1 ends with the sentence: “one representative experiment is shown”: over how many repeats? Do the authors refer to all of the panels and experiments, or a single one? Please clarify.
Response 2: The number of replicates has been added to the figure legend: n=3 for panels a and c and n=2 for panel b.
Comment 3: Figure 1: in panel 1b, the authors show an increase of nuclear Hsp110 upon TMZ treatment. Yet in panel 1d, while the total amount of Hsp110 remains constant between TMZ treated and untreated cells, the amount of immunopurified nuclear Hsp110 appears very similar between both conditions. If this discrepancy is due to the quantities loaded on the gel for WB experiments in Panel 1d, the authors should state so in their manuscript. Otherwise, how do the authors explain this observation?
response 3: For the immunoprecipitation experiment, the quantity of proteins loaded was adjusted in order to obtain a comparable amount of immunoprecipitated HSP110 from the nuclear-enriched fractions of UT and TMZ-treated cells. We have clarified the Materials and Methods section and the figure legend.
Comment 4: Proteomics analysis: please indicate the number of replicates. If the experiment was only performed once, it can only be considered as preliminary. Or at least, some of the proteomics results presented in Table 1 (differential enrichment of various proteins in TMZ treated/untreated IPs) should be validated by western blot analyses. The current good practice is to perform triplicates/quadriplicates of IPs (biological replicates, not technical ones), and interpret them in volcano plots showing fold change of enrichment vs. p-value. In addition, the proteomics data must be deposited to a recognised database such as the ProteomeXchange Consortium / PRIDE. In Table 1, what do the different grey levels correspond to? Please add the color-coding meaning to the legend for more clarity.
Response 4: HSP110 Immunoprecipitation has been performed many times and it has been very difficult to get sufficient HSP110 proteins from nuclear fractions, particularly for untreated cell samples. We had to pool our samples to have sufficient proteins to perform the mass spectrometry analysis. The present work is focused on the recruitment of HSP110 in SWI-SNF complexes and we confirmed the interaction of HSP110 with SMARCC2.
We had some controls to supports the results of our proteomic analysis. We detected DNA-PKcs accordingly to our previous publication demonstrating the role of HSP110 in the NHEJ DNA repair pathway. The interaction of HSP110 with Histone H2A, H3.1, H4, nucleolin and eEF-1α has already been reported by Wheat et al. in an article that characterized HSP-associated chaperonin network (Wheat et al. PNAS 2021, https://doi.org/10.1073/pnas.2023360118). This observation was added in the discussion, line 332-343. HSP70 and HSP110 were often found in complex, HSP110 can act as a nucleotide exchange factor and HSP70 cooperates with HSP110 to reactivate misfolded substrate. Accordingly, we detected HSP70 in our proteomic analysis. Many of the hits found in our proteome had also been identified in a proteomic analysis of HSP70 partners such as Nucleophosmin, CFA, PBRM1, ACF1, H3.1, H4, nucleolin, DNA-PKcs, ABP1, hnRNP A1, H, M, K, RNA-BP14, Rab-6A, Ran, KRas and also SWI/SNF complex subunits SMARCA4, SMARCC2, SMARCB1 and SMARCE1 (Ryu et al. PLOS Biology 2020 https://doi.org/10.1371/journal.pbio.3000606. We added the reference and discussed this observation.
Comment 5: Figure 2 a and c: Hsp110 WB lines corresponding to UT and TMZ treatments (chromatin-bound fractions) display significantly different profiles between panel in figure 2a and 2c. For instance, in panel 2a, the Hsp110 signals in UT condition increase between fraction 0.1 and 0.5, while it decreases in panel 2c. Could the authors propose an explanation about this observation? Is it because of a significant variability between replicates?
Response 5: There was some variability between experiments in the chromatin-bound fractions, probably linked to a variability in the efficacy of extracting chromatin-bound proteins. However, the observation that the amount of HSP110 in chromatin-enriched fractions increased in TMZ- or OXA-treated cells compared to those from untreated cells was highly reproducible.
In Panel 2c, due to the large number of samples, the 0.1, 0.3 and 0.5 M NaCl-chromatin-bound enriched fractions were deposited on separate gels. Therefore, it is not possible to compare the profiles of the 0.1, 0.3 and 0.5M NaCl fractions. The exposure conditions of our membrane could differ. Because the amount of HSP110 protein in the 0.5 M fraction of the “OXA” sample is very high, this may have resulted in an underestimation of the quantity of HSP110 in the untreated sample. We re-analysed our blots and selected a new exposure.
Minor comments:
- Material and methods section, Mass spectrometry analyses paragraph l. 172, please correct the first sentence: “HSP110 from nuclear-enriched fractions of control and temozolomide-treated cells, cells were lysed on ice for 20 minutes in IP buffer (1X) “
Response : the sentence has been corrected
- Strange presentation of Table 1 (words are randomly split into 2 consecutive lines. For example, the last n of nucleophosmin is on the next line
Response: We corrected the presentation of Table 1.
Discussion:
- With this study, Hsp110 appears to play a role of quaternary chaperone allowing the assembly of the SWI/SNF complex. In various cell types and compartments, this quaternary chaperone function is also performed by other molecular machines such as the PAQosome (RuvBL1/2 associated with Hsp70/90 and other helper proteins) – perhaps the authors should draw a parallel between Hsp110 and the latter?
Response: We introduced the role of HSP90 as a regulator of the assembly of macromolecular complexes in association with the PAQosome and developed our discussion (line 361-370 and 414-419).
- 356 “(…)but its’ activity, and whether its’ recruitment(…)”: both “its” do not use an apostrophe
Response: We corrected the sentence.
Round 2
Reviewer 2 Report
Comments and Suggestions for Authors
O. Pointeau and colleagues have integrated all my remarks their revised manuscript. All of my concerns have been adequately addressed. I have no further comments to make, and I believe that this work can be published in Cells in its current form.